# Moaw: Unleashing Motion Awareness for Video Diffusion Models

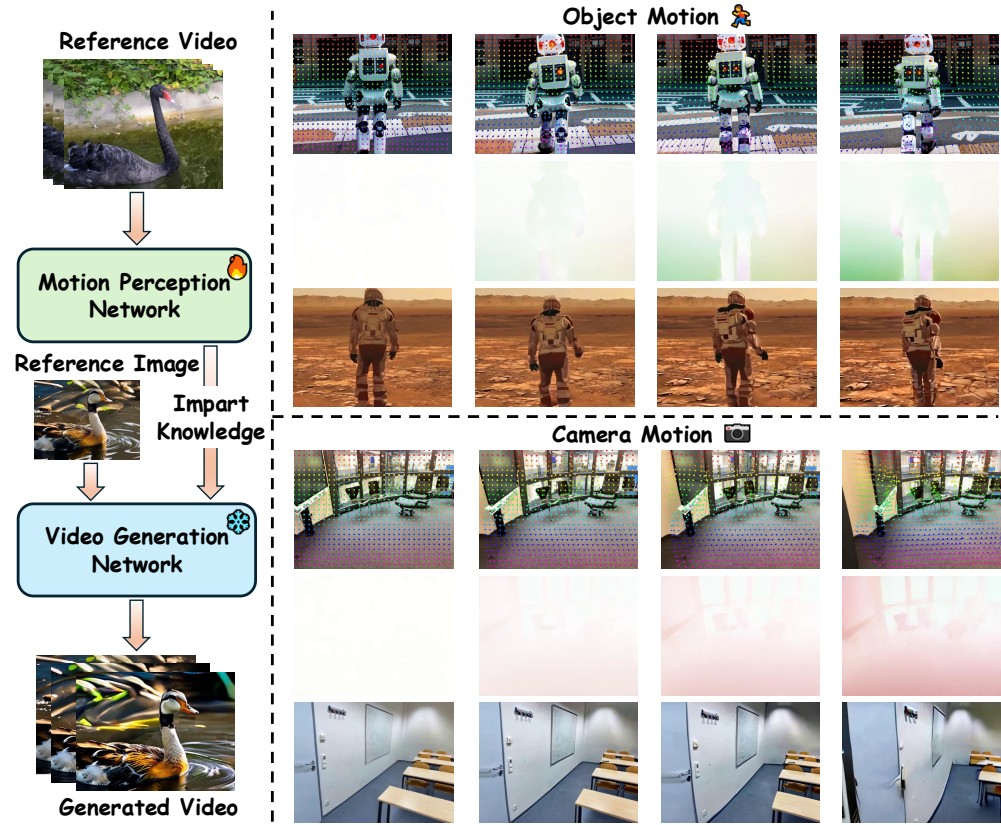

Figure 1: **Moaw** is a framework that incorporates a motion perception diffusion network and injects its features into a video generation diffusion model in a zero-shot manner to achieve motion transfer.

## Abstract

Video diffusion models, trained on large-scale datasets, naturally capture correspondences of shared features across frames. Recent works have exploited this property for tasks such as optical flow prediction and tracking in a zero-shot setting. Motivated by these findings, we investigate whether supervised training can more fully harness the tracking capability of video diffusion models. To this end, we propose Moaw, a framework that unleashes motion awareness for video diffusion models and leverages it to facilitate motion transfer. Specifically, we train a diffusion model for motion perception, shifting its modality from image-to-video generation to video-to-dense-tracking. We then construct a motion-labeled dataset to identify features that encode the strongest motion information, and inject them into a structurally identical video generation model. Owing to the homogeneity between the two networks, these features can be naturally adapted in a zero-shot manner, enabling motion transfer without additional adapters. Our work provides a new paradigm for bridging generative modeling and motion understanding, paving the way for more unified and controllable video learning frameworks.

# 1 INTRODUCTION

Video diffusion models have advanced rapidly in recent years, with systems such as Stable Video Diffusion (Blattmann et al., 2023), CogVideoX (Yang et al., 2024) and Wan (Wan et al., 2025) pushing the frontier of high-quality, temporally consistent video generation. These models demonstrate the growing scalability and versatility of diffusion-based approaches, underscoring their potential as a foundation for controllable and general-purpose video synthesis.

A central challenge in video generation is enabling precise and flexible control. Such control signals can originate from multiple modalities, including text, images (Yang et al., 2024), audio (Ruan et al., 2023), or video (Gu et al., 2025), and may consist of a single modality or a combination of several. Among these options, using a reference video as the control signal is the most challenging, especially when the generated video is required to faithfully preserve the motion information of the reference. The difficulty further increases if the first frame of the generated video is also specified by the user, since this imposes additional constraints on the video generation model, leaving far less flexibility for the model to explore and express motion creatively. Current approaches to video motion transfer can be broadly divided into two categories. The first line of work builds upon pretrained video diffusion models by analyzing their attention maps and enforcing that the generated video shares similar inter-frame attention patterns with the reference video (Pondaven et al., 2025) (Zhang et al., 2025). The second line adopts self-supervised learning, where the model learns to treat tracking information as prompts, thereby generating videos consistent with the given tracking cues (Geng et al., 2025) (Gu et al., 2025). However, the first class of methods does not fully exploit the intrinsic tracking capability of video diffusion models and typically requires iterative optimization, resulting in low efficiency. The second class, on the other hand, relies on sparse tracking prompts, which provide insufficiently dense control signals.

Prior studies observe that video diffusion models trained on large-scale video datasets, naturally capture correspondences across frames (Jiang et al., 2025). This property has been leveraged for zero-shot optical flow estimation and point trajectory prediction (Nam et al., 2025) (Kim et al., 2025). Inspired by these findings, we propose **Moaw** (Figure 1), which fully unleashes the **Mo**tion **aw**areness of video diffusion models and injects motion features into a video generation model to achieve zero-shot motion transfer. Specifically, we first train a diffusion model for motion perception, shifting its modality from image-to-video generation to video-to-dense-tracking. We then construct a motion-labeled dataset to identify the features' motion information. Finally, we inject these features directly into corresponding layers of a structurally identical video generation model. Benefited from the homogeneity between the two diffusion networks, the injected features can be naturally adapted in a zero-shot manner, enabling effective motion transfer without additional adapters.

Compared to point-tracking methods, our motion perception diffusion achieves comparable accuracy in occlusion handling while offering significantly faster inference speed. In comparison with existing motion transfer methods, our approach evaluates motion transfer accuracy through tracking and improves the End-Point Error (EPE) metric by approximately a factor of two. Our main contributions can be summarized as follows:

- We propose Moaw, a framework that unleashes the motion awareness of video diffusion models and enables zero-shot motion transfer without the need for additional adapters.
- We design a motion perception diffusion network that takes a video as input and predicts its 3D dense trajectory, shifting the modality from video generation to motion perception.
- We construct a motion-labeled dataset and conduct a systematic feature analysis to identify which layers encode the strongest motion information. The selected features are then injected into a video generation diffusion model in a zero-shot manner, enabling accurate and efficient motion transfer.

# 2 RELATED WORK

**Point Tracking.** The line of work on point tracking originally emerged from the task of optical flow estimation. Optical flow estimates dense correspondences between adjacent video frames. Classical variational methods (Horn & Schunck, 1980; Brox et al., 2004) struggle with large motion and occlusion, while deep models like FlowNet (Ilg et al., 2017) and RAFT (Teed & Deng, 2020)

significantly improved short-term estimation using correlation volumes and iterative refinement. However, these models are limited to short-term predictions and prone to error accumulation in longer sequences. Sparse tracking methods follow selected keypoints over long durations. Particle Video (Sand & Teller, 2008) and TAP-Vid (Doersch et al., 2022) demonstrated long-term tracking but only for a limited number of points. Recent methods (Harley et al., 2022) improve temporal coherence with attention or adaptive windows but remain sparse and computationally inefficient for dense tracking. Dense point tracking extends correspondence to all pixels across time, but few methods scale well due to high memory cost and temporal drift. 3D dense tracking further adds depth estimation, as explored in SceneTracker (Wang et al., 2024) and SpatialTracker (Xiao et al., 2024), but often relies on expensive modules like cross-track attention. Our work enables efficient dense 3D trajectory prediction via latent diffusion, combining long-term temporal modeling with spatial scalability. A representative work is DELTA (Ngo et al., 2024), which introduces an efficient framework for dense 3D tracking using coarse-to-fine attention and a transformer-based upsampler. However, existing 3D dense tracking methods, such as DELTA (Ngo et al., 2024), require an off-the-shelf depth estimation model to first predict per-frame depth maps, which are then fed into the tracker alongside the RGB video. This two-stage design not only introduces additional computational overhead, but also leads to long inference times due to complex attention mechanisms. In contrast, our motion perception diffusion part is an end-to-end solution that does not rely on precomputed depth; instead, it learns depth-aware representations internally.

**Video Motion Transfer.** Video motion transfer aims to synthesize new videos that follow the motion of a reference sequence. Representative approaches differ mainly in the choice of motion signal and the injection mechanism. Motion Prompting (Geng et al., 2025) conditions generation on spatio-temporal point trajectories, enabling camera/object control and motion transfer via trajectory-guided conditioning. MotionEditor (Tu et al., 2024) performs motion editing by adding a content-aware motion adapter to ControlNet with attention injection to preserve appearance while altering motion. MoTrans (Li et al., 2024) targets customized transfer in T2V models by decoupling appearance and motion and injecting motion-specific embeddings for few-video adaptation. DiTFlow (Pondaven et al., 2025) is tailored to Diffusion Transformers, extracting Attention Motion Flow from cross-frame attention of a frozen DiT to guide training-free, zero-shot motion transfer. Diffusion as Shader (DaS) (Gu et al., 2025) argues for 3D-aware control using 3D tracking videos, yielding more globally consistent geometry and temporally coherent motion transfer within a unified architecture. Unlike these methods, our approach leverages a motion perception diffusion network trained to predict dense 3D trajectories, from which motion-sensitive features are identified. These features are then injected into a video generation diffusion model in a zero-shot manner, enabling accurate and efficient motion transfer without the need for additional adapters.

## 3 PROPOSED APPROACH

In this section, we present the overall design of our proposed approach, as shown in Figure 2. In Section 3.1, we train a diffusion network for motion perception, which takes a video as input and predicts its 3D trajectory. In Section 3.2, we construct a motion-labeled dataset and use it to analyze the features of the motion perception diffusion model, identifying those that encode the strongest motion information. After that, we inject the selected features into a structurally identical video generation diffusion model in a zero-shot manner, thereby enabling motion transfer.

### 3.1 TAMING VIDEO DIFFUSION MODELS FOR MOTION PERCEPTION

**Motion Perception Problem Setup.** Our objective is to shift the modality of video diffusion models from image-to-video generation to video-to-dense-tracking. Specifically, given an input RGB video $V \in \mathbb{R}^{T \times H \times W \times 3}$, where $T$ denotes the number of frames, and $H$ and $W$ represent the spatial resolution of each frame, the target is to output a dense 3D trajectory tensor $P \in \mathbb{R}^{T \times H \times W \times 4}$.

This tensor $P$ contains the spatio-temporal information of $H \times W$ trajectories, with each trajectory originating from the corresponding pixel in the first frame. At each time step $t$, the element $P[t, i, j]$ is a 4-dimensional vector:

$$P[t, i, j] = (u_t, v_t, d_t, o_t),$$

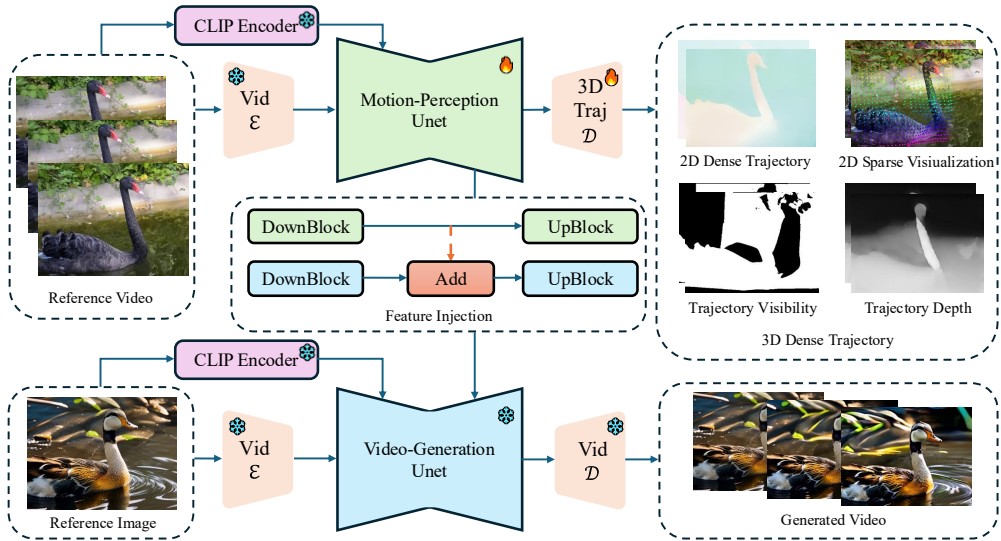

Figure 2: **Overall pipeline.** Our pipeline consists of two stages. The first stage is motion perception from a reference video. We encode the reference video into the latent space using the SVD video encoder, then sample Gaussian noise and concatenate it with the reference video latent along the channel dimension. A U-Net is then employed to predict the clean latent (without noise), which is finally decoded into a 3D dense trajectory. The second stage is video generation. Here, we take the features extracted from the first stage and inject them into a fixed SVD U-Net. Conditioned on an input image, the model generates a new video whose motion resembles that of the reference video.

where $(u_t, v_t)$ denotes the 2D image coordinates of the pixel that originated at location $(u_0, v_0) = (i, j)$ in the first frame, $d_t$ is the estimated depth of the point at time $t$, and $o_t \in \{0, 1\}$ is a visibility flag indicating whether the point is visible at time $t$.

**Latent Space Encoding.** We adopt the Latent Diffusion Model (LDM) framework (Salimans & Ho, 2022), operating in a compact latent space via a VAE originally used by Stable Video Diffusion (SVD) to encode/decode video frames (Blattmann et al., 2023), as shown in Figure 2. Prior work shows this VAE reconstructs depth with negligible error (Hu et al., 2024; Ke et al., 2024), and we observe similarly minimal distortion for both depth maps and binary visibility maps, indicating the latent space preserves key structure needed for dense prediction.

To model 3D dense trajectories, we decompose them into motion $(u, v)$, depth $d$, and visibility $o$. Their latents are $z^{uv}$, $z^d$, and $z^o$, concatenated as

$$z^{\text{traj}} = \text{cat}(z^{uv}, z^d, z^o).$$

Since diffusion expects image-like inputs, we further transform the motion component $(u, v)$ into an image-form representation before processing.

**Color 2D Dense Displacement.** Inspired by optical-flow visualization, we first convert the 2D dense trajectory to a displacement field by subtracting the first frame (a fixed grid set by resolution). Given $\Delta(u, v) \in \mathbb{R}^{H \times W \times 2}$, we compute per-pixel $\theta = \text{atan2}(v, u)$ and $r = \sqrt{u^2 + v^2}$, then normalize

$$\theta_{\text{norm}} = \frac{\theta + 2\pi \, \mathbf{1}_{\{\theta < 0\}}}{2\pi}, \qquad r_{\text{norm}} = \text{clip}\left(\frac{r}{\sqrt{H^2 + W^2}}, 0, 1\right).$$

We map each vector to HSV with direction in hue and magnitude in saturation, using

$$\text{HSV} = \big(\theta_{\text{norm}}, r_{\text{norm}}, 0.5 + 0.5(1 - r_{\text{norm}})\big),$$

and convert to RGB. This yields smooth, direction-aware visualizations suitable as RGB-like inputs for diffusion, and the process is reversible (HSV↔RGB details in the appendix).

**VAE Finetuning.** Although the colored 2D displacement field $\tilde{x}$ appears visually indistinguishable after reconstruction by the pretrained VAE from Stable Video Diffusion (SVD), minor RGB reconstruction errors may lead to significant deviations when recovering the displacement field due to the nonlinear polar-to-Cartesian mapping.

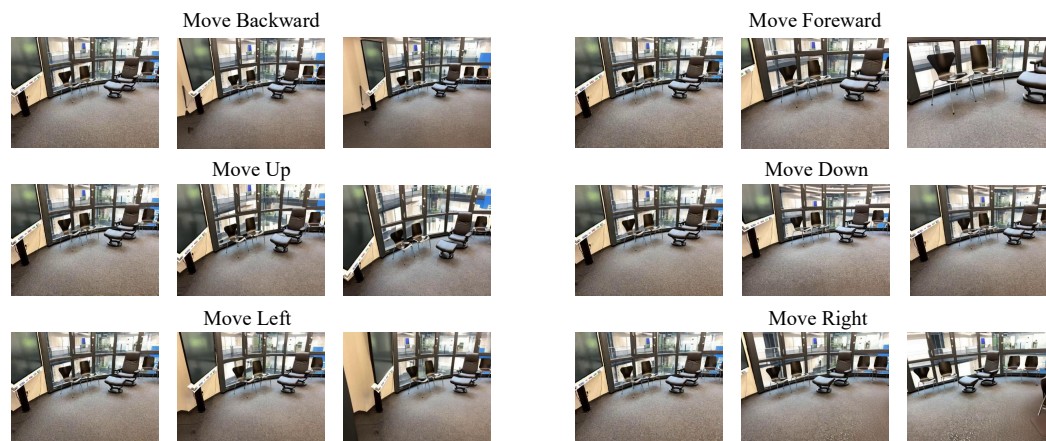

Figure 3: **Motion-labeled video dataset.** We construct a motion-labeled video dataset using images from ScanNet++ and the Stable Virtual Camera model. For each image, we generate six motion sequences, with each video consisting of 20 frames.

To mitigate this issue, we fine-tune the VAE using the colored displacement data. Specifically, we freeze the encoder and only fine-tune the decoder, ensuring that the latent space distribution of the colored displacement remains aligned with that of other modalities. This design helps maintain latent consistency and facilitates joint modeling in the diffusion process.

**Latent Diffusion Over 3D Trajectory with Video Conditioning.** Unlike SVD, which conditions on individual images, our method conditions on the entire video sequence. We utilize a latent diffusion model (LDM) that operates in the latent space of the 3D dense trajectory, guided by the full video context. Specifically, we employ a spatio-temporal U-Net to model the denoising process in latent space, incorporating video conditioning in two complementary ways.

First, we apply **latent concatenation**: the input video is encoded frame-by-frame into latent features using a pretrained VAE, and these framewise latents are concatenated with the corresponding noisy 3D trajectory latents along the channel dimension to form the U-Net input. Second, we introduce **cross-attention conditioning**: high-level clip features are extracted from the input video via a lightweight frame encoder, and injected into the mid-block of the U-Net through cross-attention. This design allows the model to leverage both local and global context during the denoising process.

During training, the U-Net learns to predict the clean latent trajectory $z^{\text{traj}}$ from its noisy version $z_\epsilon^{\text{traj}} \sim \mathcal{N}(z^{\text{traj}}, \sigma^2 \mathbf{I})$. The training objective is to minimize the denoising error under a weighted L2 loss. During inference, we sample an initial latent $z_0 \sim \mathcal{N}(0, \mathbf{I})$, and use the UNet to iteratively denoise it into a plausible trajectory latent guided by the video condition.

## 3.2 Zero-shot Motion Transfer

**Motion-labeled Video Dataset.** We aim to identify which features encode the strongest motion information, so that these features can later be injected into the generative model to enable zero-shot motion transfer video generation. During the motion perception stage, it is straightforward to extract intermediate features; however, in the absence of motion annotations, these features cannot be effectively analyzed. To address this limitation, we construct a motion-labeled video dataset comprising six distinct types of camera motion (left, right, up, down, forward, backward). Specifically, we select 160 images from the ScanNet++ dataset (Yeshwanth et al., 2023) and, for each image, employ a camera-control video generation model (Zhou et al., 2025) to synthesize videos corresponding to the six motion categories, as shown in Figure 3.

**Feature Analysis.** When selecting which features to analyze, we take inspiration from the architecture of ControlNet. ControlNet introduces a neural network structure to control diffusion models by adding additional conditions: it attaches a trainable copy outside the original diffusion U-Net and injects the residual features from the down blocks together with the features from the mid block into the original U-Net. Following this design, we also select all residual features from the down blocks and the features from the mid block (a total of $3 \times 4 + 1 = 13$) as the objects of our analysis.

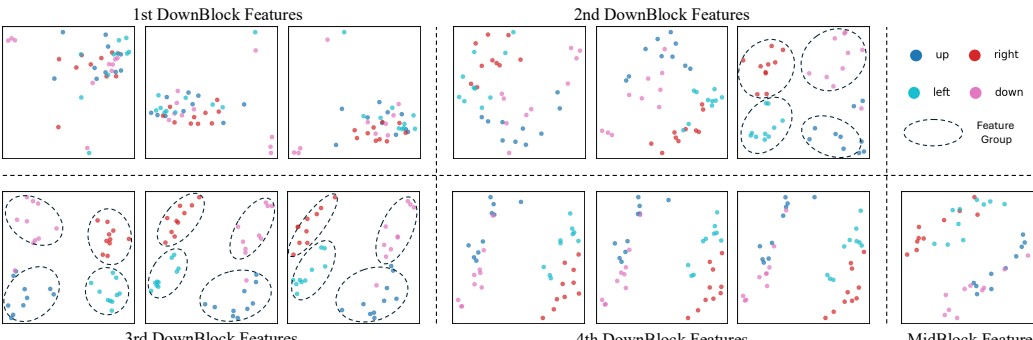

Figure 4: **PCA analysis for our motion perception video diffusion model.** We select four motion patterns, and for each pattern randomly sample ten videos, resulting in 40 samples per feature. We then apply PCA for dimensionality reduction and visualization of these features.

As illustrated in Figure 4, the first five features do not provide a clear separation of different motion types after dimensionality reduction with PCA. This suggests that these early-layer features contain relatively weak motion information and are instead dominated by low-level visual cues such as RGB appearance and texture. In contrast, starting from the sixth feature, we observe that the features extracted by our motion perception diffusion from videos with different motions begin to cluster. This indicates that these features encode strong motion information, with large differences across distinct motion types and small differences among similar ones. In particular, we find that all features from the third down block exhibit this property, suggesting that this entire block captures rich motion-related information. Although the fourth down block features display this effect less strongly, a clustering trend is still evident—likely because reducing such high-dimensional features to two dimensions inevitably results in information loss. Finally, we observe that the mid-block feature does not show the same degree of separability across motion types, implying that while intermediate representations contribute to motion encoding, they may not be sufficient on their own for reliable motion discrimination. This analysis highlights which layers of the network are most informative for motion perception and provides practical guidance for selecting features in our subsequent motion transfer experiments. Ultimately, we select the features from the third and fourth blocks as a whole, a choice that proves more beneficial for subsequent video generation.

**Zero-Shot Motion Transfer.** Based on the features selected, we proceed to zero-shot motion transfer. Following the design of ControlNet, we directly add the residual features from the third and fourth down blocks of the motion perception U-Net to the corresponding positions in the video generation U-Net, as illustrated in the Figure 2. Regardless of the number of inference steps performed by the video generation U-Net, the injected features are consistently taken from the motion perception U-Net at the first denoising step, ensuring stable and transferable motion control.

An important factor enabling this transfer is that the two U-Nets share a nearly identical architecture, and the parameters of the motion perception U-Net can be traced back to the video generation U-Net from which it was originally derived. As a result, even though the prediction modality has been changed—from video generation to motion perception—the extracted features remain structurally compatible with the video generation U-Net. This architectural alignment allows the video generation U-Net to naturally adapt to the injected features without any additional adapters, treating them as control signals that directly influence the generative process. Consequently, the model is able to synthesize videos whose motion faithfully reflects the perceived motion of the reference video, thereby achieving zero-shot motion transfer.

## 4 EXPERIMENTS

### 4.1 DENSE POINT TRACKING

**Benchmark Datasets.** We evaluate our motion-perception network across a range of tracking benchmarks on both 2D and 3D domains, including long-range optical flow and dense 3D tracking.

**Long-range 2D Optical Flow.** We adopt the CVO benchmark (Wu et al., 2023), which includes two original subsets: *CVO-Clean* and *CVO-Final*, the latter incorporating motion blur. Each subset

Table 1: **2D Long-range optical flow results** on CVO (Wu et al., 2023).

| Methods | CVO-Clean (7 frames) | | CVO-Final (7 frames) | | CVO-Extended (48 frames) | |
|---|---|---|---|---|---|---|
| | EPE↓ (*all/vis/occ*) | IoU↑ | EPE ↓ (*all/vis/occ*) | IoU↑ | EPE↓ (*all/vis/occ*) | IoU↑ |
| RAFT (Teed & Deng, 2020) | 2.48 / 1.40 / 7.42 | 57.6 | 2.63 / 1.57 / 7.50 | 56.7 | 21.80 / 15.4 / 33.4 | 65.0 |
| MFT (Neoral et al., 2024) | 2.91 / 1.39 / 9.93 | 19.4 | 3.16 / 1.56 / 10.3 | 19.5 | 21.40 / 9.20 / 41.8 | 37.6 |
| TAPIR (Doersch et al., 2023) | 3.80 / 1.49 / 14.7 | 73.5 | 4.19 / 1.86 / 15.3 | 72.4 | 19.8 / 4.74 / 42.5 | 68.4 |
| CoTracker (Karaev et al., 2024) | 1.51 / 0.88 / 4.57 | 75.5 | 1.52 / 0.93 / 4.38 | 75.3 | 5.20 / 3.84 / 7.70 | 70.4 |
| DOT (Le Moing et al., 2024) | 1.29 / 0.72 / 4.03 | 80.4 | 1.34 / 0.80 / 3.99 | 80.4 | 4.98 / 3.59 / 7.17 | 71.1 |
| SceneTracker (Wang et al., 2024) | 4.40 / 3.44 / 9.47 | - | 4.61 / 3.70 / 9.62 | - | 11.5 / 8.49 / 17.0 | - |
| SpatialTracker (Xiao et al., 2024) | 1.84 / 1.32 / 4.72 | 68.5 | 1.88 / 1.37 / 4.68 | 68.1 | 5.53 / 4.18 / 8.68 | 66.6 |
| DELTA (2D) (Ngo et al., 2024) | **0.89 / 0.46 / 2.96** | 78.3 | **0.97 / 0.55 / 2.96** | 77.7 | **3.63** / 2.67 / **5.24** | 71.6 |
| DELTA (3D) (Ngo et al., 2024) | 0.94 / 0.51 / 2.97 | 78.7 | 1.03 / 0.61 / 3.03 | 78.3 | 3.67 / **2.64** / 5.30 | 70.1 |
| **Ours** | 8.25 / 7.61 / 11.61 | **95.8** | 8.20 / 7.59 / 11.43 | **95.8** | 39.89 / 36.31 / 48.55 | **77.9** |

Table 2: **Dense 3D tracking results** on the Kubric-3D dataset.

| Methods | Kubric-3D (24 frames) | | | Time (min) |
|---|---|---|---|---|
| | AJ↑ | APD$_{3D}$ ↑ | OA↑ | |
| SpatialTracker (Xiao et al., 2024) | 42.7 | 51.6 | 96.5 | 9.00 |
| SceneTracker (Wang et al., 2024) | – | 65.5 | – | 5.00 |
| DELTA (Ngo et al., 2024) | **81.6** | **88.6** | **96.6** | 0.5 |
| **Ours** | 43.7 | 54.7 | 94.6 | **0.12** |

contains around 500 videos of 7 frames rendered at 60 FPS. Following DOT (Le Moing et al., 2024), we also include the *CVO-Extended* split, which consists of 500 videos with 48 frames rendered at 24 FPS. All CVO subsets are annotated with dense long-range 2D optical flow and occlusion masks. Although our motion perception diffusion is designed as a framework for 3D dense trajectory estimation, we can directly extract the 2D components of the predicted trajectories for evaluation and comparison with standard baselines.

Results are shown in Table 1. Quantitatively, our model achieves slightly higher end-point error (EPE) than specialized optical flow methods. However, it significantly outperforms baselines in visibility prediction accuracy, as measured by the Intersection over Union (IoU) metric. Qualitative visualizations further suggest that the slightly higher EPE does not noticeably degrade the visual quality of the predicted motion fields, which remain temporally consistent and spatially coherent.

**Dense 3D Tracking.** We evaluate dense trajectory reconstruction on the Kubric dataset (Greff et al., 2022), which contains 143 synthetic videos, each consisting of 24 frames with ground-truth 3D dense trajectory annotations. Results are shown in Table 2.

Our method achieves state-of-the-art inference efficiency, significantly outperforming prior methods in runtime while maintaining competitive accuracy. Compared to DELTA, our model is approximately 4× faster at inference time, while achieving comparable performance in Occlusion Accuracy (OA). Although our overall scores are slightly lower than DELTA on other metrics, the trade-off in speed makes our method more suitable for practical deployment. When compared to SpatialTracker, our model achieves significantly better results on both Average Jaccard (AJ) and Average Points within Threshold (APD$_{3D}$), while being approximately 75× faster. These results demonstrate the high computational efficiency and scalability of our approach without sacrificing core tracking quality.

### 4.2 Video Motion Transfer

For the motion transfer task, we conduct a comparison with Diffusion As Shader (DAS). From our constructed dataset, we randomly select 20 videos for each motion category (120 videos in total) as reference videos. For each reference video, we further sample a random image from ScanNet++ as the condition image, and then run both DAS and our model 120 times, obtaining 120 pairs of reference and generated videos for each method. To evaluate performance, we employ the DELTA model as a fair "judge", which compares the 2D dense tracking between the reference and generated videos and computes the End-Point Error (EPE).

As shown in Table 3, our method significantly outperforms DAS across nearly all motion categories in terms of EPE. In particular, our approach achieves improvements on challenging camera motions

Table 3: **Motion Transfer Comparison**

| Method | EPE ↓ | | | | | | | Time(min) ↓ |
| | down | up | left | right | forward | backward | mean | mean |
| --- | --- | --- | --- | --- | --- | --- | --- | --- |
| DAS | 40.30 | 29.15 | 32.95 | 42.60 | 16.20 | **12.06** | 28.88 | 6.5 |
| **Ours** | **14.74** | **17.87** | **16.04** | **19.70** | **12.36** | 14.14 | **15.81** | **0.5** |

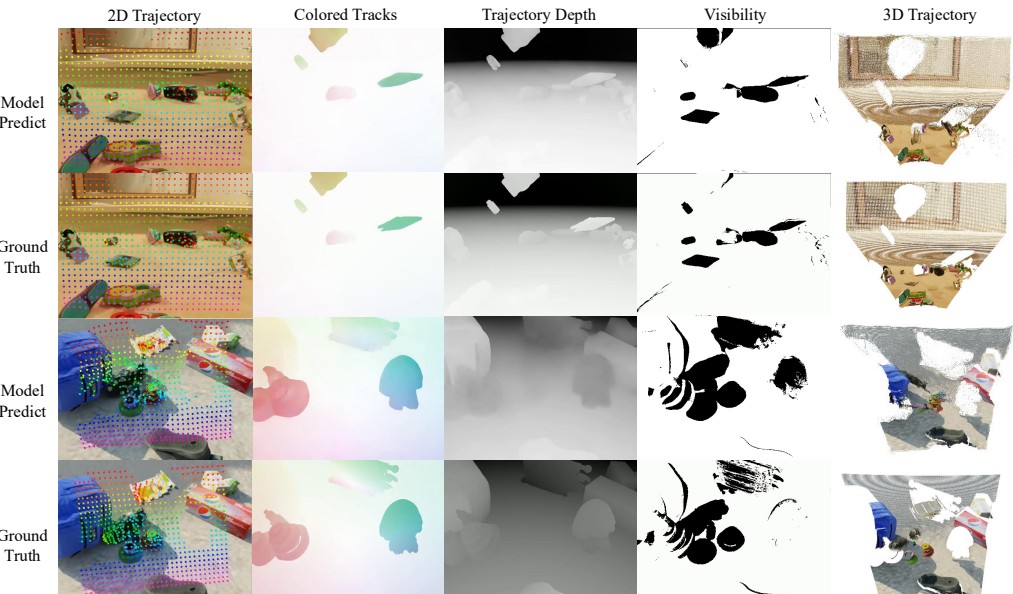

Figure 5: **Visualizations.** It shows qualitative comparisons between model predictions and ground truth for two samples. The top two rows correspond to examples from the training set, while the bottom two rows are from the validation set. Each sample includes 10 subfigures: the first row shows model predictions, and the second row shows the corresponding ground truth. From left to right, the columns display: (1) visualizations of subsampled 2D dense trajectories, (2) predicted colored tracks, (3) trajectory depth, (4) visibility maps, and (5) visualizations of 3D trajectoy.

such as "down" and "right", reducing the error by more than half. Although DAS performs slightly better on the backward motion, our model still yields a competitive result while maintaining superior performance overall. Our method reduces the mean EPE from 28.88 to 15.81, demonstrating an enhancement in capturing motion consistency between reference and generated videos.

In this speed evaluation, we perform only a single inference step for the tracking component, without completing the full denoising and decoding process to obtain a motion map. Instead, we directly use the features extracted at this first step to guide our motion transfer. All runtime measurements reported in the table are obtained on an A800 GPU. Our method's inference time per video is only 0.5 minutes, compared to 6.5 minutes for DAS, highlighting its effectiveness. These results validate the capability of our framework to transfer motion more accurately and efficiently, while preserving temporal coherence across diverse motion patterns.

## 4.3 EXPERIMENTAL ANALYSIS

**Visualizations.** We visualize the outputs of our motion perception diffusion model and compare them with the ground truth, as shown in Figure 5. Although our model lags behind methods such as DELTA in quantitative metrics, the generated trajectories capture overall motion trends and remain visually consistent with the reference. These results indicate that, despite room for improvement, our model shows strong potential for motion perception and transfer tasks.

**Ablation Study.** To validate the effectiveness of our selected features, we conduct an ablation study, as shown in Figure 6. We compare multiple sources and combinations of features under a reference video with a move-backward motion pattern. Without any injected features (no condition), the generated video exhibits an incorrect motion, drifting toward the lower-left corner. When replacing our motion perception diffusion with SVD and selecting features from the 3rd and 4th blocks, the

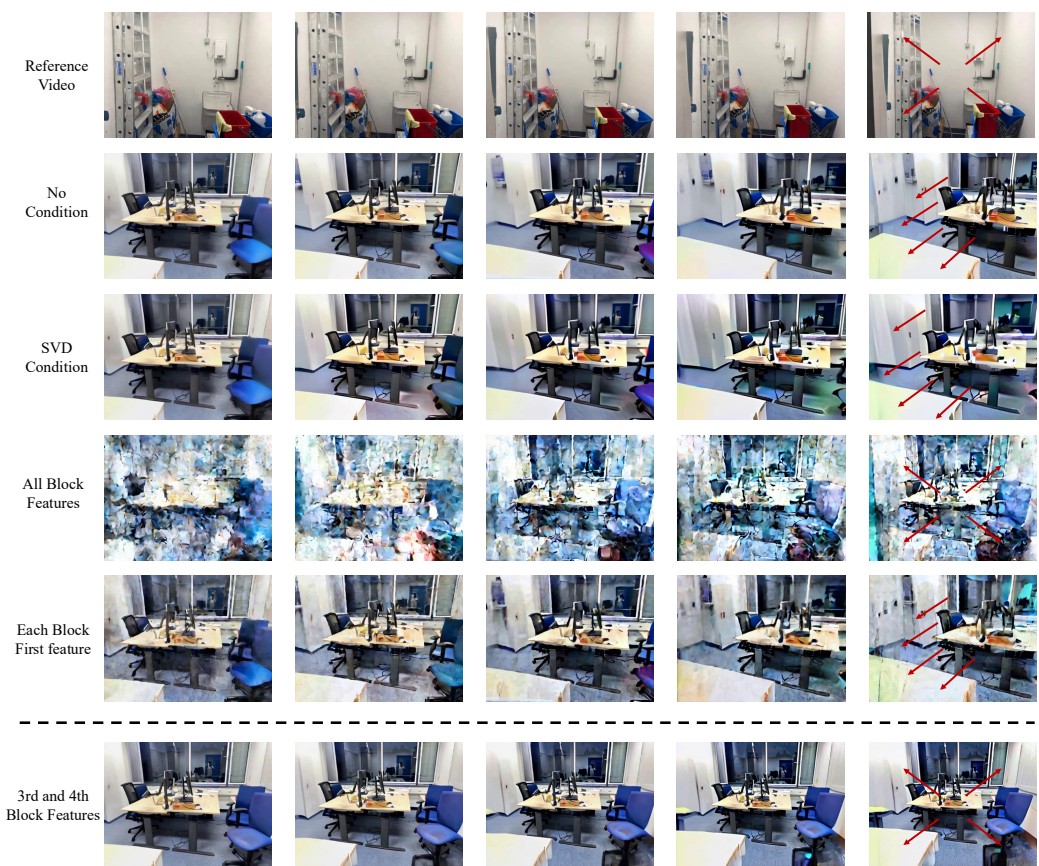

Figure 6: **Ablation study on different feature selections.** We compare different sources and combinations of features. The baselines include using no condition, replacing our motion perception diffusion with SVD and selecting features from the 3rd and 4th blocks, using all features from the motion perception diffusion, taking only the first feature from each block, and finally, selecting the 3rd and 4th block features from the motion perception diffusion. In the visualizations, the red arrow in the last frame indicates the overall motion trend of the generated video.

result resembles the no-condition case but appears even blurrier. Using all features from the motion perception diffusion produces motion consistent with the reference video, but the output is noticeably blurred. When taking only the first feature from each block, the generated motion is again incorrect, and the video lacks clarity. Finally, when selecting the 3rd and 4th block features from the motion perception diffusion, the generated motion pattern closely matches that of the reference video and remains relatively sharp. These results confirm that features from the 3rd and 4th blocks are the most effective signals for motion transfer, consistent with our analysis in Section 3.2.

## 5 CONCLUSION

In this work, we have presented Moaw to incorporate motion perception diffusion and inject its features into a video generation diffusion model in a zero-shot manner to achieve motion transfer. By training a motion perception diffusion network with the constructed motion-labeled dataset, we have identified motion-sensitive features and demonstrated how they can be directly injected into a video generation diffusion model to enable zero-shot motion transfer. Moaw achieves strong motion consistency with improved efficiency, surpassing point-tracking methods in speed and outperforming prior motion transfer approaches in accuracy. Beyond empirical gains, this work highlights the under-explored potential of video diffusion models as both generative and motion-perceptive systems. We believe Moaw provides a new paradigm for bridging generative modeling and motion understanding, paving the way toward more unified, controllable, and versatile video learning.

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

# A APPENDIX

## A.1 RGB-HSV CONVERSION DETAILS

To enable differentiable and interpretable color encoding of the displacement field, we rely on the transformation between the RGB and HSV color spaces.

Given a pixel with normalized RGB components $R, G, B \in [0, 1]$, we define:

$$C_{\max} = \max(R, G, B), \quad C_{\min} = \min(R, G, B), \quad \Delta = C_{\max} - C_{\min}.$$

Then, the HSV components are computed as follows:

- **Hue** $H \in [0, 1]$:

$$H = \begin{cases} 0, & \Delta = 0 \\ \frac{(G-B)}{\Delta} \mod 6, & C_{\max} = R \\ \frac{(B-R)}{\Delta} + 2, & C_{\max} = G \\ \frac{(R-G)}{\Delta} + 4, & C_{\max} = B \end{cases} \quad \text{and} \quad H \leftarrow \frac{H}{6}.$$

- **Saturation** $S \in [0, 1]$:

$$S = \begin{cases} 0, & C_{\max} = 0 \\ \frac{\Delta}{C_{\max}}, & C_{\max} \neq 0 \end{cases}$$

- **Value** $V \in [0, 1]$:

$$V = C_{\max}.$$

The inverse mapping from HSV back to RGB proceeds as follows. Given $H \in [0, 1]$, $S \in [0, 1]$, and $V \in [0, 1]$, define:

$$C = V \cdot S, \quad X = C \cdot (1 - |(6H \mod 2) - 1|), \quad m = V - C.$$

Let $H' = 6H \in [0, 6]$. Then the RGB components before adding the offset $m$ are:

$$(R', G', B') = \begin{cases} (C, X, 0), & 0 \leq H' < 1 \\ (X, C, 0), & 1 \leq H' < 2 \\ (0, C, X), & 2 \leq H' < 3 \\ (0, X, C), & 3 \leq H' < 4 \\ (X, 0, C), & 4 \leq H' < 5 \\ (C, 0, X), & 5 \leq H' < 6 \end{cases}$$

$$(R, G, B) = (R' + m, \ G' + m, \ B' + m).$$

This transformation ensures that directional information (angle) and magnitude (displacement length) can be encoded and decoded smoothly through color.

## A.2 3D METRICS

$$\text{APD}_{3D} = \frac{1}{V} \sum_{i,t} v_t^i \cdot \mathbf{1} \left( \left\| \hat{P}_t^i - P_t^i \right\| < \delta_{3D}(P_t^i) \right),$$

where:

- $\hat{P}_t^i \in \mathbb{R}^3$: the predicted 3D position of the $i$-th point at time $t$,
- $P_t^i \in \mathbb{R}^3$: the ground truth 3D position of the same point,
- $v_t^i \in \{0, 1\}$: ground truth visibility flag (1 if visible, 0 if occluded),
- $\delta_{3D}(P_t^i) = Z(P_t^i) \cdot \delta_{2D}/f$: the adaptive 3D distance threshold derived by unprojecting a pixel threshold $\delta_{2D}$ using the depth $Z(P_t^i)$ and camera focal length $f$,

- $V = \sum_{i,t} v_t^i$: the total number of visible points,

- $\mathbf{1}(\cdot)$: the indicator function, equal to 1 if the condition holds, otherwise 0.

$$\mathrm{OA} = \frac{1}{P} \sum_{i,t} \mathbf{1}\left(\hat{v}_t^i = v_t^i\right),$$

where:

- $\hat{v}_t^i \in \{0,1\}$: the predicted visibility flag for point $i$ at time $t$, where 1 means visible and 0 means occluded,

- $v_t^i \in \{0,1\}$: the ground truth visibility flag,

- $\mathbf{1}(\cdot)$: the indicator function that returns 1 if the predicted and ground truth flags match, and 0 otherwise,

- $P$: the total number of point-time pairs, i.e., all evaluated $(i,t)$ across the sequence.

$$\mathrm{AJ}_{3D} = \frac{\sum_{i,t} v_t^i \, \hat{v}_t^i \, \alpha_t^i}{\sum_{i,t} v_t^i + \sum_{i,t}\left((1 - v_t^i)\hat{v}_t^i + v_t^i \, \hat{v}_t^i(1 - \alpha_t^i)\right)},$$

where:

- $\hat{v}_t^i \in \{0,1\}$: predicted visibility flag for the $i$-th point at time $t$,

- $\alpha_t^i = \mathbf{1}\left(\left\|\hat{P}_t^i - P_t^i\right\| < \delta_{3D}(P_t^i)\right)$: whether the prediction is within the adaptive threshold,

- Numerator: counts the number of **true positives** — visible points that are predicted visible and have low error,

- Denominator: adds **false positives** (points predicted visible but are occluded or inaccurate), and **false negatives** (points that are visible but predicted either as occluded or inaccurate).

### A.3 3D TRAJECTORY VAE DETAILS

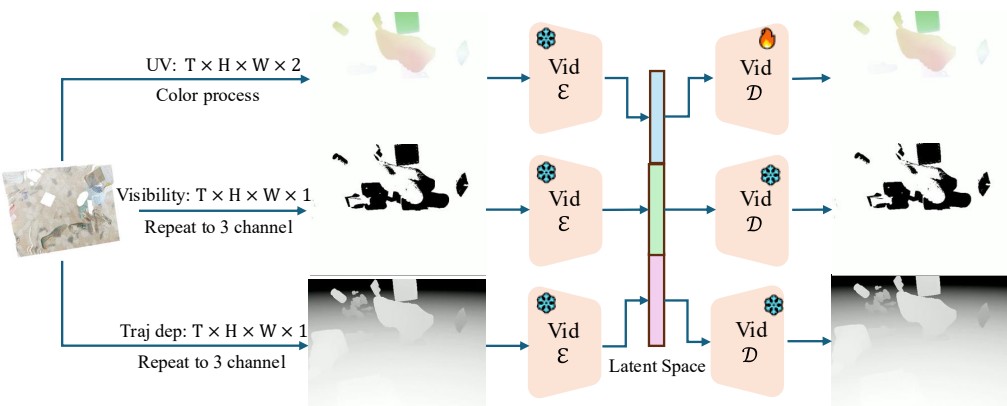

Figure 7: **3D Trajectory encoder and decoder.** We decompose the 3D trajectories into separate components and encode them using the same pretrained encoder, ensuring latent space consistency with the input video. For the decoding stage, our colored tracks decoder is fine-tuned to achieve higher reconstruction accuracy.

## B ETHICS STATEMENT

This work adheres to the ICLR Code of Ethics. In this study, no human subjects or animal experimentation was involved. All datasets used, including our constructed Motion-labed dataset, were

sourced in compliance with relevant usage guidelines, ensuring no violation of privacy. We have taken care to avoid any biases or discriminatory outcomes in our research process. No personally identifiable information was used, and no experiments were conducted that could raise privacy or security concerns. We are committed to maintaining transparency and integrity throughout the research process.

## C    REPRODUCIBILITY STATEMENT

We have made every effort to ensure that the results presented in this paper are reproducible. All code and datasets have been made publicly available in an anonymous repository to facilitate replication and verification. The experimental setup, including training steps, model configurations, and hardware details, is described in detail in the paper. We have also provided a full description of our Moaw framework, to assist others in reproducing our experiments.

Additionally, tracking datasets in the paper, such as Kubric, are publicly available, ensuring consistent and reproducible evaluation results.

We believe these measures will enable other researchers to reproduce our work and further advance the field.

## D    LLM USAGE

Large Language Models (LLMs) were used to aid in the writing and polishing of the manuscript. Specifically, we used an LLM to assist in refining the language, improving readability, and ensuring clarity in various sections of the paper. The model helped with tasks such as sentence rephrasing, grammar checking, and enhancing the overall flow of the text.

It is important to note that the LLM was not involved in the ideation, research methodology, or experimental design. All research concepts, ideas, and analyses were developed and conducted by the authors. The contributions of the LLM were solely focused on improving the linguistic quality of the paper, with no involvement in the scientific content or data analysis.

The authors take full responsibility for the content of the manuscript, including any text generated or polished by the LLM. We have ensured that the LLM-generated text adheres to ethical guidelines and does not contribute to plagiarism or scientific misconduct.

