# OpenReview forum: "Moaw: Unleashing Motion Awareness for Video Diffusion Models"
_ICLR.cc/2026/Conference — Submitted to ICLR 2026_

### Official Review · Reviewer_bNk2 · 2025-10-31

**Soundness:** 2
**Presentation:** 3
**Contribution:** 2
**Rating:** 2
**Confidence:** 3

**Summary:**

This paper introduces Moaw, a framework designed to endow video diffusion models with motion awareness for zero-shot motion transfer. The core idea is to train a motion perception diffusion network that converts videos into 3D dense trajectories—effectively shifting the modality from image-to-video generation to video-to-dense-tracking. These features are then injected into a structurally identical video generation diffusion model, enabling motion transfer without any additional adapters or fine-tuning.

**Strengths:**

1. The work presents a novel cross-modal paradigm: using a diffusion model trained for motion perception to guide another diffusion model for generation, in a zero-shot manner.

2. The motivation, problem formulation, and experimental flow are logical and easy to follow.

3.  The work suggests a direction for motion-aware generative modeling, potentially influencing future controllable video generation systems.

**Weaknesses:**

1. While Moaw achieves strong IoU and OA metrics, its EPE on 2D optical flow benchmarks is significantly higher than standard trackers as shown in Table 1. This raises concerns about whether the motion perception module truly captures fine-scale motion or relies on global coherence.

2. The motion-labeled dataset includes only six synthetic camera motions, which may not generalize to object-centric or nonrigid motions common in real videos.

3. Feature injection is performed only at early blocks (3rd/4th down-blocks). Although ablations justify this choice, it would be helpful to show whether combining multiple hierarchical features (e.g., multi-scale fusion) further improves results.

**Questions:**

1. How sensitive is the downstream motion transfer performance to the accuracy of the motion perception diffusion? For instance, would improved tracking (lower EPE) yield proportionally better motion transfer?

2. Does the motion perception model trained with CVO benchmark dataset?

3. Are the injected features normalized or projected before fusion? Could feature misalignment between networks (even with shared architecture) affect stability?

4. To get the feature for Figure 4, how the three types of feature was splitted? equally split to three parts?

5. Can the framework handle nonrigid or articulated motion (e.g., human movement)? If so, could the authors discuss potential dataset extensions or adaptations?

6. The model achieves impressive speedups; is this mainly due to using fewer denoising steps, or architectural simplification? Could the same acceleration techniques be applied to competing models for fairer runtime comparison?

---

### Official Review · Reviewer_MxVg · 2025-11-01

**Soundness:** 2
**Presentation:** 2
**Contribution:** 2
**Rating:** 6
**Confidence:** 3

**Summary:**

1. The authors convert an img-to-video diffusion model like SVD into a video-to-dense 3d tracking model using a similar diffusion based approach as the base model.
2. The authors create a motion labelled dataset using ScanNet++ images and an off-the-shelf camera controlled video generation model.
3. The authors then use these motion annotations to analyse the features from different blocks of this dense 3d tracking denoiser model and perform PCA to find which features are maximally separated based on motion.
4. The authors then add the selected features from a reference video exhibiting a certain motion to the layer representations in the pre-trained SVD model to zero-shot transfer the reference motion to the new generated video.
5. The authors quantitatively evaluate the tracking and motion following using existing benchmarks and the created benchmark respectively.

**Strengths:**

1. The authors present a novel method to zero-shot transfer motion from a reference video to a newly generated video.
2. The authors also create a dense video 3d point tracking model in the process which trades tracking accuracy for inference latency compared to existing approaches.
3. The authors demonstrate qualitatively and quantitatively the effectiveness of their motion transfer approach.

**Weaknesses:**

1. It would be interesting to see, how the motion transfer method generalises to more complex motions, for example generated using RecamMaster [1], using the same selected features as in the paper.
2. The selected feature ablation needs some quantitative results as well.
3. One interesting analysis to do would be to separate fore-ground object motion and camera motion and see if they can be transferred independent of each other.
4. It would be very relevant and timely to try out this same method on newer DiT based video gen models like CogX [2]



[1] Bai et al., “ReCamMaster: Camera-Controlled Generative Rendering from A Single Video,” arXiv preprint arXiv:2503.11647, 2025. https://arxiv.org/abs/2503.11647

[2] Yang et al., “CogVideoX: Text-to-Video Diffusion Models with An Expert Transformer,” arXiv preprint arXiv:2408.06072, 2025. https://arxiv.org/abs/2408.06072

**Questions:**

see weaknesses.

---

### Official Review · Reviewer_taCx · 2025-11-01

**Soundness:** 2
**Presentation:** 3
**Contribution:** 2
**Rating:** 2
**Confidence:** 4

**Summary:**

This paper proposes Moaw, a two-stage framework for video motion transfer. The authors first train a motion perception diffusion model that predicts dense 3D trajectories from a reference video. They then conduct feature analysis using a constructed camera-motion dataset to identify layers that encode strong motion signals. These motion-aware features are injected into a structurally identical video generation model in a zero-shot fashion, enabling motion transfer without additional adapters. The approach is evaluated on 3D tracking benchmarks (Kubric, CVO) and on camera-motion transfer tasks, showing improved zero-shot EPE compared to Diffusion-as-Shader (DAS).

**Strengths:**

- Using motion-visualization maps (color-mapped trajectories) as input features helps capture coarse motion while reducing appearance dependence.
- Converting trajectories to a colormap and aligning them with the VAE latent space is a clever design choice.
- PCA-based feature analysis is a reasonable attempt at interpretability. The finding that mid-level blocks encode the most discriminative motion information is intuitive and aligns with observations from ControlNet-style architectures.

**Weaknesses:**

- The authors argue that no adapter is needed, but the method requires training a full diffusion UNet on dense 3D trajectories with the same parameter count as the generation UNet. Compared to lightweight adapter methods, this approach may be less efficient overall. A rigorous comparison of training/inference cost against adapter-based baselines is missing, making the “adapter-free” claim feel incomplete.

- The motion-labeled dataset consists only of six controlled camera motions on static ScanNet++ scenes. This restricts the study to camera-motion transfer rather than general motion transfer.

- Implementation details are missing, especially for training schedule and dataset scale for trajectory training

- In Table1 and 2, the motion perception model performs much worse than other tracking baselines. More exploration is needed to improve tracking quality before using these features for motion transfer.

**Questions:**

- Did you evaluate the method on complex object motion, not just camera motion in static scenes?
- After VAE finetuning, how much reconstruction error improves compared to the original VAE?
- Why is depth included? Could motion be represented using only x, y, and visibility?
- What training data and schedule were used for the motion perception model?

---

### Official Review · Reviewer_yiqk · 2025-11-01

**Soundness:** 3
**Presentation:** 3
**Contribution:** 3
**Rating:** 4
**Confidence:** 3

**Summary:**

This paper proposes a framework that investigates the tracking ability of video diffusion models.  It first trains a motion perception model to predict the dense trajectory.  After that, the proposed method identifies the motion-aware features with a new dataset, and injects these features into identically the same video diffusion models. This paper evaluates long-range 2D optical flow and dense 3D tracking tasks.

**Strengths:**

1. The idea of training a video diffusion model to predict dense trajectories is interesting.
2. This paper designs a motion-labeled video dataset for the analysis of motion-aware features is insightful.
3. The paper is well-written and easy to follow.

**Weaknesses:**

1. Despite the interesting design, the proposed method achieves worse results than the baselines, as shown in Table 1 and Table 2. This limits the application of the proposed method.

2. This paper highlights the point tracking task, but lacks one important benchmark, TAP-Vid [1]. How about the tracking performance compared to non-diffusion methods such as CoTracker3 on TAP-Vid?

[1] TAP-Vid: A Benchmark for Tracking Any Point in a Video

3. This paper evaluates videos of no more than 48 frames. However, in real applications, videos could be much longer than that. How does the method track longer videos?

4. According to line 89, this paper claims comparable accuracy in occlusion handling. However, the only result is OA reported in Table 2, which is lower than both baseline methods. This makes the claim not solid.

5. Figure 5 and Figure 6 show some visualizations of the proposed method. However, these are limited to scenarios with objects. How about real-world applications, such as those involving human motions?

6. How about performing a similar PCA analysis using features from the original video diffusion models in Figure 4? Will the trend of selecting layers be the same as in the motion perception model?

**Questions:**

Please refer to the weaknesses.

---

### Meta-Review · Area_Chair_F2Pt · 2026-01-07

**Summary:**

The paper initially received an overall score of 2,2,4,6. Most reviewers find the proposed cross-modal paradigm—using a diffusion model trained for motion perception to guide another diffusion model for generation in a zero-shot manner—interesting. However, all reviewers raised concerns regarding the work, including inferior quantitative performance compared to strong baselines, incomplete or unconvincing experimental validation, limited motion diversity and generalization, unclear scalability and real-world applicability, the need for clarification on efficiency and the “adapter-free” claim, missing analyses and ablations, as well as insufficient implementation details. The authors did not provide a rebuttal to address these concerns.

**Reviewer Concerns:**

The authors did not provide a rebuttal to address the reviewers’ concerns.

**Reviewer Scores:**

As the authors did not provide a rebuttal, it is likely that the original scores will be maintained or possibly lowered.

---

### Decision · Program_Chairs · 2026-01-26

Reject